# Analysis of *De Novo* Mutations in Sporadic Cardiomyopathies Emphasizes Their Clinical Relevance and Points to Novel Candidate Genes

**DOI:** 10.3390/jcm9020370

**Published:** 2020-01-29

**Authors:** Maria Franaszczyk, Grazyna Truszkowska, Przemyslaw Chmielewski, Malgorzata Rydzanicz, Joanna Kosinska, Tomasz Rywik, Anna Biernacka, Mateusz Spiewak, Grazyna Kostrzewa, Malgorzata Stepien-Wojno, Piotr Stawinski, Maria Bilinska, Pawel Krajewski, Tomasz Zielinski, Anna Lutynska, Zofia T. Bilinska, Rafal Ploski

**Affiliations:** 1Molecular Biology Laboratory, Department of Medical Biology, Institute of Cardiology, 04-628 Warsaw, Poland; m.fran@wp.pl (M.F.);; 2Unit for Screening Studies in Inherited Cardiovascular Diseases, Institute of Cardiology, 04-628 Warsaw, Poland; 3Department of Medical Genetics, Medical University of Warsaw, 02-106 Warsaw, Poland; 4Department of Heart Failure and Transplantology, Institute of Cardiology, 04-628 Warsaw, Poland; 5Postgraduate School of Molecular Medicine, Medical University of Warsaw, 02-091 Warsaw, Poland; 6Magnetic Resonance Unit, Department of Radiology, Institute of Cardiology, 04-628 Warsaw, Poland; 7Department of Forensic Medicine, Medical University of Warsaw, 02-007 Warsaw, Poland; 8Department of Arrhythmia, Institute of Cardiology, 04-628 Warsaw, Poland; 9Department of Medical Biology, Institute of Cardiology, 04-628 Warsaw, Poland

**Keywords:** cardiomyopathy, genetic, de novo mutation, novel genes

## Abstract

The vast majority of cardiomyopathies have an autosomal dominant inheritance; hence, genetic testing is typically offered to patients with a positive family history. A de novo mutation is a new germline mutation not inherited from either parent. The purpose of our study was to search for de novo mutations in patients with cardiomyopathy and no evidence of the disease in the family. Using next-generation sequencing, we analyzed cardiomyopathy genes in 12 probands. In 8 (66.7%), we found de novo variants in known cardiomyopathy genes (*TTN*, *DSP*, *SCN5A*, *TNNC1*, *TPM1*, *CRYAB*, *MYH7*). In the remaining probands, the analysis was extended to whole exome sequencing in a trio (proband and parents). We found de novo variants in genes that, so far, were not associated with any disease (*TRIB3*, *SLC2A6*), a possible disease-causing biallelic genotype (*APOBEC* gene family), and a de novo mosaic variant without strong evidence of pathogenicity (*UNC45A*). The high prevalence of de novo mutations emphasizes that genetic screening is also indicated in cases of sporadic cardiomyopathy. Moreover, we have identified novel cardiomyopathy candidate genes that are likely to affect immunological function and/or reaction to stress that could be especially relevant in patients with disease onset associated with infection/infestation.

## 1. Introduction

Cardiomyopathies are a group of diseases defined as structural and functional abnormalities in the cardiac muscle that cannot be explained by abnormal loading conditions [1]. Cardiomyopathies have a well-established genetic background and their genetic heterogeneity is characterized by the overlapping of disease genes for different clinical types [2]. The majority of cardiomyopathies have an autosomal dominant inheritance pattern. Therefore, genetic testing is typically offered to patients with a positive family history, whereas sporadic cases may mistakenly be classified as acquired without considering the potential role of de novo mutations. This is particularly important considering pediatric cases with no family history.

A de novo mutation is a new germline mutation caused by mutagenesis occurring in parental gametes, thus not inherited from either parent. Such a genetic variant occurs for the first time in one family member, thus forming the first generation of carriers, who can then transmit this new variant to their offspring.

It should be noted that the majority of cardiomyopathies causing de novo mutations have been reported in syndromic cases such as those caused by variants in *DSP* (Naxos–Carvajal syndrome [3], erythrokeratodermia-cardiomyopathy syndrome [4]), *LAMP2* (Danon disease [5]), *PRKAG2* (glycogen storage disease of heart [6,7]), *RAF1* (Noonan syndrome [8]), *TAZ* (Bart syndrome [9]), *RRAGC* (syndromic fetal dilated cardiomyopathy [10]), and *LMNA* (atypical progeroid syndrome and dilated cardiomyopathy [11]). Data on de novo mutations in non-syndromic cardiomyopathies are limited and include variants in *ACTC* [12], *MYH7* [13,14,15,16], *TNNI3* [17], *TNNT2* [18,19], and *TPM1* [20,21].

The purpose of our study was to search for de novo mutations in patients with cardiomyopathy and no evidence of the disease in the family after a detailed clinical evaluation of proband-parent trios and, if available, other relatives.

## 2. Materials and Methods

Criteria for inclusion in the study required a pattern of the disease in the given family consistent with the possibility of a de novo mutation. Clinical examination of probands with cardiomyopathy included 12-lead electrocardiography (ECG), two-dimensional Doppler echocardiography, coronary angiography or coronary computed tomography angiography (CTA), serum creatine phosphokinase (CPK) and, if clinically indicated, cardiac magnetic resonance (CMR). The criteria for parents were clinically examined and healthy (based on normal non-invasive cardiac examination – ECG, echocardiogram), both living or at least with well-documented non-cardiac death and available genetic material.

Dilated cardiomyopathy was defined according to European Society of Cardiology (ESC) criteria: Left ventricular ejection fraction (LVEF) below 45% and left ventricular end-diastolic diameter (LVEDD) >117% of the predicted value, corrected for age and body surface area [1].

Left ventricular noncompaction was diagnosed based on CMR study with the ratio of noncompacted to compacted myocardium greater than 2.3 during diastole on long-axis cine images [22].

Hypertrophic cardiomyopathy was defined as left ventricular hypertrophy in the absence of loading conditions, sufficient to account for the observed degree of hypertrophy, with a maximal left ventricular wall thickness ≥15 mm in one or more myocardial segments [1].

Diagnostic criteria for restrictive cardiomyopathy were restrictive ventricular physiology in the presence of normal or reduced diastolic volumes (of one or both ventricles), normal or reduced systolic volumes, and normal ventricular wall thickness [1].

None of the probands had hypertension, diabetes, or clinically significant atherosclerosis. For parents, these morbidities (when the criteria for cardiomyopathy were not met) were not considered an exclusion criterion.

The study was approved by the local Bioethics Committee of the Institute of Cardiology with approval numbers 1276 and 1451. All patients examined in the study signed written informed consent in accordance with the Declaration of Helsinki. Genetic testing and blood sampling were offered to all probands and relatives who agreed to participate in the study. All proband-parent trios have undergone both paternity and maternity testing.

DNA was extracted from the peripheral blood by phenol extraction or the salting-out method. In the probands, we performed next-generation sequencing including whole exome sequencing (*n* = 10), TruSight One panel (*n* = 1), and TruSight Cardio panel (*n* = 1), (Illumina, San Diego, CA, USA). Whole exome sequencing (WES) libraries were constructed using the TruSeq Exome Enrichment Kit (Illumina), Nextera Rapid Capture Exome Kit (Illumina), SeqCap EZ MedExome kit (Roche, Basel, Switzerland), and SureSelectXT Human All Exon v5 (Agilent Technologies, Santa Clara, CA, USA) according to the manufacturers’ instructions and analyzed as described previously [23]. TruSight One and TruSight Cardio panel sequencing were performed according to the manufacturer’s instructions. All libraries were pair-end sequenced on HiSeq 1500 or MiSeq (Illumina). A minimum of 20× coverage was obtained for the mean of 81%±13 of target regions. The results were visualized using Integrative Genomics Viewer v.2.3.81 (IGV, http://software.broadinstitute.org/software/igv/).

Parenthood was confirmed by the Forensic Medicine Department of the Medical University of Warsaw. DNA samples were short tandem repeats (STR) genotyped using PowerPlex Fusion 6C (Promega, Madison, WI, USA).

Baseline analysis of next generation sequencing (NGS) results consisted of searching for genetic variants with very-low-frequency (<0.001) and high-bioinformatic-pathogenicity prediction scores with special regard to phenotypically plausible genes. The frequencies of variants were derived from gnomAD (http://gnomad.broadinstitute.org), NHLBI GO Exome Sequencing Project (ESP) 6500 (https://esp.gs.washington.edu/drupal), and an in-house database of >1000 Polish subjects examined by WES. For the bioinformatic prediction scores, we used data summarized in VarSome (https://varsome.com). The clinical significance of the variants was based on ClinVar (https://www.ncbi.nlm.nih.gov/clinvar).

If a potentially damaging variant in one of the known genes associated with cardiomyopathies was found in the proband, instead of performing WES for a trio, a more cost-effective approach was used to confirm or exclude the presence of variants of interest identified by NGS. We performed Sanger sequencing in a trio setting (proband-parent) and, if available, in other relatives. For Sanger sequencing, we used the 3500xL Genetic Analyzer (Life Technologies, Carlsbad, CA, USA) and BigDye Terminator v3.1 Cycle Sequencing Kit (Life Technologies) following the manufacturer’s instructions. Chromatograms were analyzed using Variant Reporter 1.1 (Life Technologies). Finding a de novo mutation in a cardiomyopathy-associated gene ended the diagnostic process.

In the remaining cases, with no candidate variant found using the strategy described above, WES was also carried out for the proband’s parents. Further analysis was performed as described above (including bioinformatic analysis and variant Sanger sequencing) with focus on rare variants present in the proband but absent in the parents. Additionally, copy number variants (CNV) analysis was performed using the CNVkit [24] with a reference population of around 30 samples.

## 3. Results

We identified 12 probands with sporadic cardiomyopathy who fulfilled the abovementioned criteria. The pedigrees are shown in corresponding Figure 1, Figure 2, and Appendix A, and clinical characteristics are given in Appendix A. Ten patients had dilated cardiomyopathy (DCM), reflecting a relatively high frequency of sporadic cases in this condition (families FD01-10). Two DCM patients additionally had left ventricular noncompaction (LVNC). One patient had an atypical course of hypertrophic cardiomyopathy (HCM) with restrictive features (family FH1) and suspicion of storage disease, and one fulfilled the criteria for restrictive cardiomyopathy (RCM) (family FR1).

### 3.1. De Novo Mutations in Known Cardiomyopathy Genes

In 8/12 probands (66.7%), we found likely causative variants in known cardiomyopathy genes (*TTN*, *DSP*, *SCN5A*, *TNNC1*, *TPM1*, *CRYAB*, and *MYH7*), which were confirmed as de novo (six DCM, one HCM, and one RCM case, Table 1, Appendix A). Considering only DCM, the prevalence of de novo mutations in the known genes was 60% (6/10). Genetic findings and genotype–phenotype correlations are described in detail below (see also Table 1 and Appendix A).

#### 3.1.1. Family FD02

In the proband, we found a de novo missense mutation p.Trp976Leu/c.2927G>T in the *TTN* gene (titin, MIM #188840). The mutation was absent in the proband’s parents and two sisters but inherited by her affected son. The variant has high deleteriousness scores (seven pathogenic predictions vs. no benign predictions) and was not found in gnomAD. Moreover, mutation in the same amino acid (p.Trp976Arg, rs267607155) was described by Gerull et al. [25] as segregating with DCM in a three-generation large kindred, and its protein-damaging effect was confirmed in a functional study [26].

The 34-year-old female proband was referred following diagnosis of DCM in her son. She was in New York Heart Association (NYHA) class II, in sinus rhythm, and, on the echocardiogram, had a dilated spherical left ventricle (LVEDD of 70 mm), LVEF of 25%, and restrictive mitral inflow pattern. No significant arrhythmia was found on the 24 hour Holter ECG. She had a history of recurrent tonsillitis. At the age of 30, she gave vaginal birth to a male child, in whom DCM was diagnosed at the age of three, following chickenpox. No other relatives had any cardiac abnormalities.

#### 3.1.2. Family FD03

In the proband, we found a de novo missense mutation p.Glu290Lys/c.868G>A in the *DSP* gene (desmoplakin, MIM #125647). The variant has high deleteriousness scores (nine pathogenic predictions vs. no benign predictions) and is absent from gnomAD. The same variant was described in a patient with arrhythmogenic right ventricular cardiomyopathy (ARVC), repolarization abnormalities, and family history, although no functional or segregation studies were performed [27].

The female proband started experiencing palpitations at the age of 26 y, and was diagnosed with DCM at 28 y. CMR showed impaired left ventricle (LV) function (LVEF 37%), late gadolinium enhancement (LGE), and normal right ventricle. Repeated Holter ECGs revealed frequent and complex ventricular arrhythmia including bursts of non-sustained ventricular tachycardia (nsVT), and a cardioverter-defibrillator (ICD) was implanted at the age of 30 y. The patient has a history of thyrotoxicosis due to autoimmune thyroid nodular disease that could aggravate her symptoms and was vigorously treated with thyrostatics. However, the complex ventricular arrhythmia and impaired left ventricle function also persisted during euthyreosis.

#### 3.1.3. Family FD04

In the proband, we found a de novo missense mutation p.Glu1548Gln/c.4642G>C in the *SCN5A* gene (sodium voltage-gated channel alpha subunit 5, MIM #600163). The variant has high deleteriousness scores (nine pathogenic predictions vs. no benign predictions) and was not found in gnomAD. Another variant in the same amino acid (p.Glu1548Lys/c.4642G>A) was described in three unrelated patients with Brugada syndrome [28].

The male proband was diagnosed because of palpitations at the age of 15 y. Since then, a variety of ventricular and supraventricular arrhythmias were identified, which led to radiofrequency ablation (RFA) of ventricular arrhythmia at the age of 17 and 22 y, and cavotricuspid isthmus-dependent atrial flutter ablation at the age of 22 y. At 23 y, his left ventricular function is mildly depressed.

#### 3.1.4. Family FD06

In the proband, we found a de novo missense mutation p.Glu94Val/c.281A>T in the *TNNC1* gene (troponin C1, slow skeletal and cardiac type, MIM #191040). The variant has high deleteriousness scores (eight pathogenic predictions vs. one benign prediction) and was not found in gnomAD. Another variant in the same amino acid (p.Glu94Ala/c.281A>C) was described in a patient with LVNC with onset at 4 mo and no family history who underwent heart transplant [29].

The 20-year-old male proband started having decreased exercise tolerance at 13 y. At 16 y, the patient suffered cardiogenic shock, had a dilated left ventricle of 75 mm and LVEF of 15%, and received a heart transplant after a bridge with a biventricular assist device (BiVAD) for three months.

#### 3.1.5. Family FD09

In the proband, two missense candidate mutations were found: p.Ile201Thr/c.602T>C in the *MYH7* gene (myosin heavy chain 7, MIM #160760) and p.Lys205Arg/c.614A>G in the *TPM1* gene (tropomyosin 1, MIM #191010). The *MYH7* variant has ambiguous deleteriousness scores (two pathogenic predictions vs. four benign predictions) while the *TPM1* variant has high deleteriousness scores (six pathogenic predictions vs. one benign prediction). Neither variant was found in gnomAD. The family study revealed that the *MYH7* variant arose de novo, while the *TPM1* variant was inherited from the proband’s asymptomatic father. The *MYH7* variant has also been described as “likely pathogenic” in a DCM patient [30].

The 19-year-old male proband was diagnosed with DCM and LVNC after echocardiographic and CMR study performed due to palpitations. LVEF was 26%, LVEDD was 62 mm, and the ratio of noncompacted to compacted myocardial layer was 3.2:1. The patient received standard heart failure treatment, and at 24 y, his left ventricle function remained reduced (LVEF 37%).

#### 3.1.6. Family FD10

In the proband, we found a de novo missense mutation p.Thr40Met/c.119C>T in the *CRYAB* gene (alpha B-crystallin, MIM #123590). The variant was inherited by the proband’s daughter. The variant has ambiguous deleteriousness scores (three pathogenic predictions vs. four benign predictions) and was found in gnomAD (f = 0.00001073).

The 44-year-old male proband suffered from an ischemic stroke at the age of 36 y and, at that time, was diagnosed with DCM, LVNC, paroxysmal atrial fibrillation, intraventricular conduction disease (QRS of 153 ms), and heart failure. Following standard heart failure treatment and with ICD implantation, he recovered substantially; however, eight months later, he had acute third-degree atrioventricular block and his ICD was upgraded to a cardiac resynchronization therapy defibrillator (CRT-D). The patient has a visual impairment with a short-sightedness of −4.5 D and astigmatism. His asymptomatic 20-year-old daughter has nonspecific ST-T changes in lateral leads on a 12-lead ECG and trabeculations in the inferolateral wall of the left ventricle, not meeting criteria for the diagnosis of noncompaction. She is also a short-sighted person.

#### 3.1.7. Family FR1

In the proband, two missense candidate mutations were found: p.Gly768Arg/c.2302G>A in *MYH7* and p.Pro1066Arg/c.3197C>G in *MYBPC3*. Both variants have high deleteriousness scores: *MYH7* has eight pathogenic predictions vs. no benign predictions, while the *MYBPC3* variant has seven pathogenic predictions vs. one benign prediction, and neither was found in gnomAD. *MYH7* p.Gly768Arg was previously described in a family with HCM [31]. The family study showed that the *MYH7* variant was de novo, while the *MYBPC3* variant was inherited from the father who did not have DCM.

The male proband was diagnosed at 28 y because he wanted to apply for a boating license, and his standard ECG was abnormal. Subsequent imaging studies including an echocardiogram, CMR, and angio-CT scan revealed a hypoplastic left ventricle with reduced diastolic volume and narrow entire aorta, and a hypertrophied right ventricle, most probably due to pulmonary hypertension. Normal or reduced diastolic volumes of the left ventricle, that resemble hypoplastic left heart, fit well into the definition of restrictive cardiomyopathy [1]. On the other hand, the narrow aorta corresponds well to the diminished size of the left ventricle. As, clinically, his disease resembled restrictive cardiomyopathy with high pulmonary hypertension, we assumed that exclusion of the other congenital abnormalities in our patient and the presence of the sarcomeric gene mutation allowed us to consider the diagnosis of restrictive cardiomyopathy in the patient.

#### 3.1.8. Family FH1

In the proband, a de novo missense mutation p.Arg453Cys/c.1357C>T in the *MYH7* gene was identified. The variant has high deleteriousness scores (eight pathogenic predictions vs. no benign predictions) and was not found in gnomAD. This *MYH7* variant has established pathogenicity – it was described in patients with HCM [32,33] and DCM [34] and associated with poor prognosis. 

The 39-year-old female proband started having exertional dyspnoea at 16 y and was diagnosed with HCM. At 32 y, she had ankle oedema following second delivery. Her heart failure symptoms progressed, and at 34 y on an echocardiogram, she was found to have asymmetric nonobstructive hypertrophic cardiomyopathy with reduced left ventricle function and prominent restrictive features. She received a heart transplant at 35 y.

### 3.2. Novel DCM Candidate Genes – Findings from WES Analysis in Trios

WES analysis was performed in four proband-parent trios (all probands with DCM) without mutation in any genes related to cardiomyopathy. In two probands, it revealed de novo variants in genes that were, so far, not associated with any disease (FD05 and FD08, Figure 1). In another proband, there was a possible disease-causing biallelic genotype (FD07, Figure 2), while, in the last proband, a de novo mosaic variant without strong evidence of pathogenicity was found (FD01, Figure 2). All variants were absent from the ClinVar database and classified with default Varsome settings as variants of uncertain significance (VUSs) by The American College of Medical Genetics and Genomics (ACMG) criteria [35]. Three of them upgraded to “likely pathogenic” when the de novo criterion was included (PS2). These findings are described in detail below (also see Table 2 and Appendix A). Technical parameters of WES in trios are shown in Appendix A. Appendix A, included in Appendix A, also shows all ultra-rare (<0.00001) variants found in probands that could contribute to the disease but were identified as inherited from parent.

#### 3.2.1. FD05

In the proband, we found a de novo missense mutation p.Gly257Ser/c.769G>A in the *TRIB3* gene (tribbles pseudokinase 3, MIM #607898) (Figure 1A). *TRIB3* has not been linked to any cardiac disease; however, the amino acid is highly conserved among distant species (Figure 1B). The variant has been found in two subjects from the gnomAD exome database (0.000008). It has seven pathogenic predictions vs. two benign predictions.

The proband was a female with severe exercise intolerance (NYHA class III) at the age of 40 y, which followed a flu-like syndrome two months earlier. She had a dilated left ventricle with moderate mitral insufficiency, global diffuse hypokinesis of LVEF 10–15%, and the coronary CTA was normal. She received a single-chamber implantable cardioverter-defibrillator (ICD-VR) and responded slowly to treatment with LVEF of 35%. At the age of 44, she is in NYHA II class heart failure.

#### 3.2.2. FD08

In the proband, we found a de novo missense mutation p.Arg283His/c.848G>A in the *SLC2A6* gene (solute carrier family 2 member 6, MIM #606813) (Figure 1A). *SLC2A6* has not been associated with any disease. The variant has ambiguous deleteriousness scores (three pathogenic predictions vs. four benign predictions), it was not found in gnomAD, and the amino acid is highly conserved (Figure 1B).

The 29-year-old female proband started having palpitations at 21 y, and at 28 y, she was diagnosed with eosinophilic myocarditis following a parasitic infestation, on the basis of blood eosinophilia and CMR examination. At 29 y, she was found to have complex ventricular arrhythmia and systolic dysfunction with LVEF 38%, and a subsequent CMR study confirmed diffuse areas of LGE. During the study, her mother’s cousin was diagnosed with cardiomyopathy but she was unavailable for the study.

#### 3.2.3. FD01

We found a potential mosaic variant (11% of reads) p.Arg633Trp/c.1897C>T in the *UNC45A* gene (Unc-45 myosin chaperone A, MIM #611219). The variant was found in 11% of 277 reads in peripheral blood and it was confirmed by Sanger sequencing (Figure 2). As the proband is deceased, study of other tissues was not performed. The variant has ambiguous predictions of deleteriousness (four pathogenic predictions vs. four benign predictions) and is relatively frequent in the South Asian population (gnomAD f = 0.00039), and also in the homozygous state (*n* = 1).

The male 36-year-old proband started having palpitations and NYHA class III exercise intolerance at 31 y. Three months earlier, he had a flu-like syndrome. He was in sinus rhythm, and diffuse left ventricular dilation and profound hypokinesis were found on the echocardiogram. Coronary angiography was normal. On the 24 hour Holter ECG, frequent ventricular ectopy was present with >20000 VEBs. The patient had repeated RF ablation of ventricular arrhythmia, with partial response. At 34 y, paroxysmal atrial fibrillation episodes occurred that led to exacerbation of heart failure symptoms. Sinus rhythm had been restored with cardioversion. He died at 36 y, probably due to an embolic episode.

#### 3.2.4. FD07

Based on copy number variant analysis confirmed by the single-nucleotide variants (SNVs) configuration from the trio analysis, we found evidence in the proband of a CNV removing the *APOBEC3B* gene (apolipoprotein B mRNA editing enzyme catalytic subunit 3B, MIM #607110). The deletion was inherited from the mother, unmasking a hemizygous p.Cys217Tyr/c.650G>A variant of paternal origin (Figure 2). The variant has high deleteriousness scores (six pathogenic predictions vs. one benign prediction) and was not found in gnomAD. The patient was also found to have haplotype IV and splice-site SV-154 in the *APOBEC3H* locus [36]. The *APOBEC3H* SV-154 variant predicts loss of function due to a loss of multiple conserved structural elements [36].

The female proband had a history of DCM, diagnosed at the age of two. The onset of the disease was associated with a bronchopulmonary infection when the dysfunction of the left ventricle was diagnosed. At 21 y, she had ablation of ventricular arrhythmia. At 24 y, she was in NYHA class II with significant left ventricular dysfunction (LVEF 30%, and LVEDD 67 mm) and frequent ventricular ectopy with few complex forms on the Holter 24 hour ECG. She refused ICD implantation. At the age of 26, she is clinically stable.

### 3.3. Parental Age

In our group, the age of fathers and mothers at the time of probands’ birth was relatively low (28.2 ± 6.3, range 21–40; and 24.6 ± 3.8, range 18–31, respectively, Appendix A). 

## 4. Discussion

The estimated rate for germline de novo single-nucleotide variants in humans is 1.0 to 1.8 × 10^−8^ per nucleotide per generation [37]. This means about 44 to 82 variants in the average human genome, with one or two (about 3%) in the coding sequence [38], although this number varies within and between families [39]. An important aspect of de novo mutations is the reproductive fitness impairment they cause. This was particularly demonstrated for neurodevelopmental diseases, in which de novo mutations are responsible for the relatively high prevalence of these disorders in the general population (reviewed in [40]). Furthermore, in the X-linked Duchenne dystrophy, the percentage of de novo mutations in affected males reaches up to 40% [41]. According to the ACMG criteria for the pathogenicity of genetic variants, the occurrence of a de novo mutation with both maternity and paternity confirmed (PS2 criterion) is exactly as strong as well-established in vitro or in vivo functional studies (PS3), and both arguments belong to the class of "strong evidence of pathogenicity" [35]. If it is de novo, the variant is usually promoted to a higher class of pathogenicity. Hence, the value of de novo mutation studies is both determining the pathogenicity of variants in known genes, as well as searching for new disease-causing genes.

We present 10 cases of de novo mutations in patients with cardiomyopathy. In 6 of 10 cases of sporadic DCM and in single cases of RCM and HCM, we found de novo mutations in genes previously associated with cardiomyopathy (*CRYAB*, *DSP*, *MYH7*, *SCN5A*, *TNNC1*, and *TTN)*. Interestingly, in all but one case, the same variants or variants at the same position have been previously associated with the disease. The exception was *CRYAB* p.Thr40Met found in a proband and his daughter who both had cardiac disease and severe myopia. In this family, the variant was inherited by a daughter who – at a relatively young age – was not a healthy person. Given the fact that cardiomyopathy has age- and gender-dependent penetrance and that the female sex often has a milder course of the disease, we assumed this to be evidence of cosegregation, thus suggesting pathogenicity of this variant. Although it is not clear whether this variant causes disease, it is noteworthy that pathogenic *CRYAB* variants are associated with cardiomyopathies/myopathies and ocular features such as cataracts (https://omim.org). However, given the limited knowledge on both the *CRYAB* pathogenic spectrum (only 11 variants are listed in ClinVar) and genetics of myopia, it is intriguing to speculate that the coexistence of dilated cardiomyopathy and myopia observed in the family FD10 may be caused by the *CRYAB* variant. In another two cases of sporadic DCM, we found de novo mutations in potential novel DCM candidate genes, *TRIB3* and *SLC2A6*. In the remaining two probands, there was a possible disease-causing biallelic genotype (*APOBEC* genes), and a de novo mosaic variant in the *UNC45A* gene without strong evidence of pathogenicity was found.

Even though a small study group does not provide reliable quantitative results, the high prevalence of de novo mutations (83% among all cardiomyopathies and 80% among DCMs) emphasizes that genetic screening is also indicated in cardiomyopathy cases without family history.

Unsurprisingly, one of the de novo variants was found in the *TTN* gene, which is the main locus for DCM. However, the variant we found was a missense, while *TTN* association with the disease applies mainly to truncating mutations. Missense *TTN* variants are abundant, due to the large size of the gene, and are generally considered benign or of uncertain significance (according to ClinVar database only <1% of *TTN* missense variants have pathogenic or likely pathogenic status). However, a de novo *TTN* missense mutation, especially when transmitted to the affected child, is likely to be pathogenic.

Another important finding is the novel *TNNC1* variant. Among troponins, *TNNC1* is the only gene whose role in cardiomyopathies is controversial, including a suggestion of the recessive model of inheritance, despite functional studies indicating the pathogenicity of monoallelic variants [42]. In our proband, the *TNNC1* variant was in the heterozygous state and arose de novo, suggesting a dominant pathogenic effect.

An important aspect of our study is the identification of novel cardiomyopathy candidate genes. It is intriguing that among probands without defects in known genes, we found plausible candidate variants (ultra-low population frequency and in silico pathogenicity prediction) in genes such as *APOBEC, TRIB3*, and *SLC2A6*, which are likely to affect immunological function and/or reaction to stress. Interestingly, all these variants were found in patients in whom the disease was associated with infection (bronchopulmonary infection, flu-like syndrome, or parasitic infestation, respectively).

The *APOBEC* genes have a well-established role in innate immunity against retroviral infection and cancers through restricting the replication of viruses and transposons. *APOBEC3B* is not indispensable, as full gene deletions exist in the general population [43]. However, a deleted *APOBEC3B* locus together with inactive *APOBEC3H* haplotypes (i.e., hapIV, which likely has no anti-viral activity at all [44]) may increase the susceptibility to infection by HTLV-1 with severe implications for disease progression [45]. Although p.Cys217Tyr is not a de novo variant, taking into account the infection-related onset of DCM in early childhood, we suggest that the specific genotype/haplotype in the *APOBEC3* family (i.e., a single copy of *APOBEC3B* damaged by the missense variant together with *APOBEC3H* hapIV) may have contributed to cardiomyopathy in this patient. Furthermore, this observation suggests a link between genetics and myocarditis, especially interesting considering that the history of myocarditis is a risk factor for dilated cardiomyopathy.

*TRIB3* is a stress-related gene that induces apoptosis during ER stress [46]. In the heart, TRIB3 has a low expression but is induced by hypoxia [47] or the transcription factor NF-kappaB, which it negatively regulates. In addition, TRIB3 sensitizes cells to TNF- and TRAIL-induced apoptosis and is involved in insulin signaling [48]. In mice, Trib3 knock-out affects mast cell function and beta cell apoptosis (http://www.informatics.jax.org/). Intriguingly, TRIB3 interacts with members of the APOBEC3 family [49].

*SLC2A6* encodes a poorly characterized glucose transporter GLUT6 with a likely function in the immune system [50]. In mice, Glut6 is upregulated by activation of T lymphocytes [51], while in humans, the GLUT6 protein concentration was shown to increase in CD4+ T cells on HIV-1 infection [52]. Recently, GLUT6 was shown to be a lysosomal transporter regulated by inflammatory stimuli and modulating glycolysis in macrophages [53] Glut6 knock-out mice had decreased resistance to LPS-induced shock [53].

The *UNC45A* gene could also be implicated in stress-related pathomechanism as it is a co-chaperone for HSP90 while another DCM-related gene, *BAG3,* is a co-chaperone for HSP70 [54]. However, the relatively high population prevalence of the variant argues against its major role as a disease cause.

The presence of a de novo mutation in a known gene is strong evidence of its pathogenicity, while in the case of an unknown gene, it only allows us to hypothesize on its possible association with the disease. Clearly, such hypotheses should be verified by functional studies and by identification of additional patients. At present, the role of the variants we found is difficult to evaluate as, to our knowledge, this is the first report of association of the respective genes with DCM. However, given the involvement of these candidate genes in immune response and an established association between DCM and inflammation, we think that our findings offer a valuable clue for further studies.

### Limitations

The estimate of de novo mutation prevalence in this study is likely overestimated due to the small number and selection of probands as, even in typical familial cardiomyopathies, the prevalence of identified mutations does not exceed 40% in DCM [55] and approx. 65% in HCM [56]. The detection of mutations was limited to the coding sequence, and the CNV analysis performed did not allow us to detect other types of structural variants such as inversions, balanced translocations, or complex rearrangements. The lack of functional analyses does not allow us to confirm the hypothetical role of the identified genes and their association with the disease.

## 5. Conclusions

The high percentage of de novo mutations observed in our cohort suggests that their occurrence in sporadic cases of cardiomyopathy is non-negligible. De novo variants detected in patients without an established cause for disease raises the possibility that genes involved in immune/stress response such as *APOBEC, TRIB3*, or *SLC2A6* could be relevant for cardiomyopathy development, especially in patients with disease onset associated with infection/infestation. However, these hypotheses need to be verified by further studies, functional analyses, and by identification of additional patients with mutations in respective genes, as the role of the variants we found is difficult to evaluate based only on their de novo origin.

## Figures and Tables

**Figure 1 jcm-09-00370-f001:**
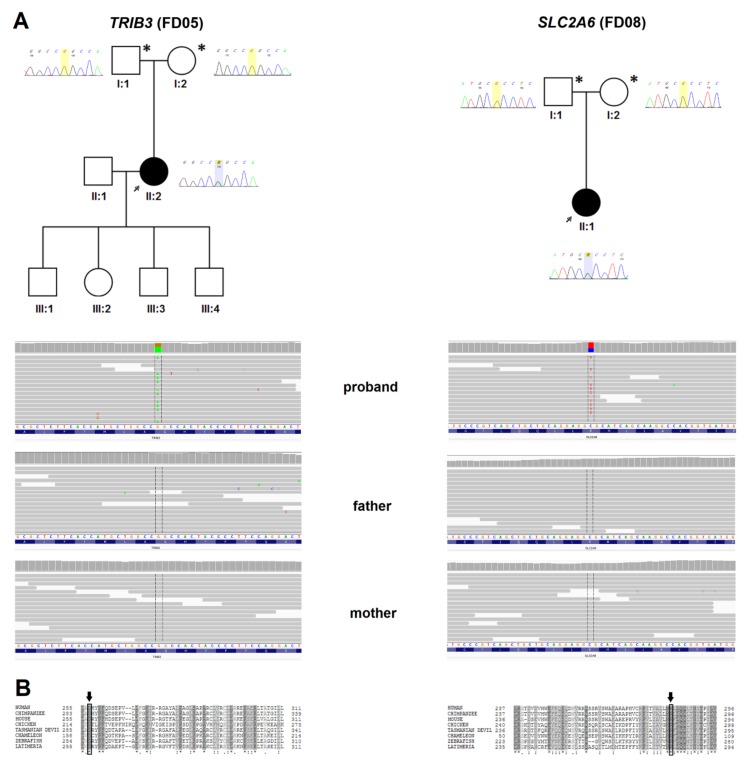
(**A**) The pedigrees of families with de novo variants in novel cardiomyopathy candidate-genes, corresponding Sanger chromatograms, and IGV views in trios. Left: *TRIB3* variant p.Gly257Ser/c.769G>A (family FD05); right: *SLC2A6* variant p.Arg283His/c.848G>A (family FD08); (**B**) Evolutionary conservation of amino acid residues altered by de novo missense mutations in probands. Aligned UniProt sequences for SLC2A6: #GTR6 (*Homo sapiens*), #H2R2P8 (*Pan troglodytes*), #A2AR26 (*Mus musculus*), #F1NAR8 (*Gallus gallus*), #G3VXJ7 (*Sarcophilus harrisii*), #H9GLX0 (*Anolis carolinensis*), #F1RB41 (*Danio rerio*), #H3AT68 (*Latimeria chalumnae*), and for TRIB3: #TRIB3 (*Homo sapiens*), #H2QJS5 (*Pan troglodytes*), #TRIB3 (*Mus musculus*), #R4GLD1 (*Gallus gallus*), #G3VRY0 (*Sarcophilus harrisii*), #H9GDK2 (*Anolis carolinensis*), #F1QCV8 (*Danio rerio*), #H3AA74 (*Latimeria chalumnae*).

**Figure 2 jcm-09-00370-f002:**
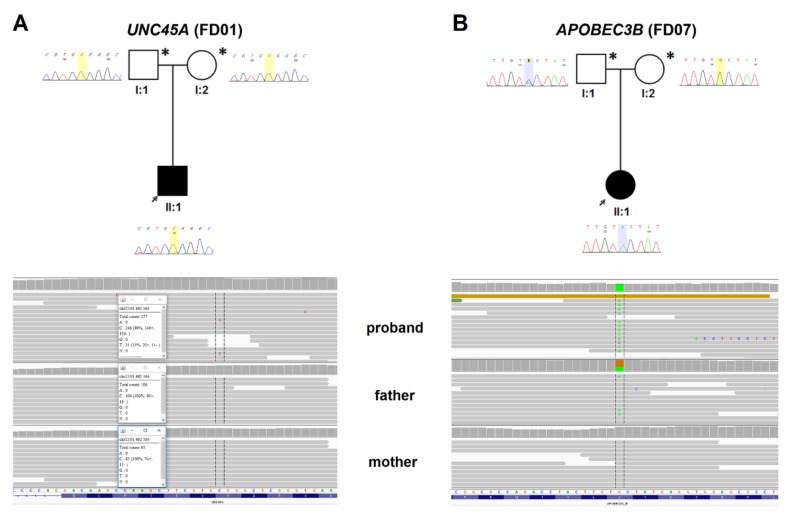
Pedigrees of families with other findings, corresponding Sanger chromatograms, and IGV views in trios. (**A**) *UNC45A* possible mosaic de novo variant p.Arg633Trp/c.1897C>T (11% of reads and corresponding lower T peak in proband’s chromatogram) (Family FD01); (**B**) *APOBEC3B* variant p.Cys217Tyr/c.650G>A (hemizygotic variant as a result of maternal copy deletion) (family FD07).

**Table 1 jcm-09-00370-t001:** Molecular characteristics of variants of interest in known cardiomyopathy genes found in probands.

Family	NGS Target	Gene/Transcript	Variant (dbSNP ID)	Genomic Coordinates (GRCh38)	Allele Frequency	ACMG Verdict (default/if de novo)	ClinVar Clinical Significance	Status
FD02	WES	*TTN/* NM_001267550.2	p.Trp976Leu/c.2927G>T ^1^ (ND)	2:178782979-C>A	0	VUS/ Likely Pathog	ND	Novel
FD03	WES	*DSP/* NM_004415.2	p.Glu290Lys/c.868G>A ^1^ (rs397516974)	6:7565449-G>A	0	VUS/ Likely Pathog	VUS (1x)	Described [27]
FD04	TSC	*SCN5A/* NM_198056.2	p.Glu1548Gln/c.4642G>C ^1^ (ND)	3:38554450-C>G	0	VUS/ Likely Pathog	ND	Novel
FD06	WES	*TNNC1/* NM_003280.2	p.Glu94Val/c.281A>T ^1^ (ND)	3:52451780-T>A	0	VUS/ Likely Pathog	ND	Novel
FD09	WES	*MYH7/* NM_000257.3	p.Ile201Thr/c.602T>C ^1^ (rs397516258)	14:23431798-A>G	0	Likely Pathog/ Pathog	Likely Pathog (3×)	Described [30]
		*TPM1/* NM_001018006.1	p.Lys205Arg/c.614A>G ^2^ (ND)	15:63061248-A>G	0	VUS/ Likely Pathog	ND	Novel
FD10	TSO	*CRYAB/* NM_001885.2	p.Thr40Met/c.119C>T ^1^ (rs782122417)	11:111911606-G>A	1.07e–05 (GnomAD)	VUS/ Likely Pathog	ND	Novel
FR1	WES	*MYH7/* NM_000257.3	p.Gly768Arg/c.2302G>A ^1^ (rs727503260)	14:23425403-C>T	0	Pathog/ Pathog	Pathog (1×)	Described [31]
		*MYBPC3/* NM_000256.3	p.Pro1066Arg/ c.3197C>G ^2^ (ND)	11:47333327-G>C	0	VUS/ Likely Pathog	ND	Novel
FH1	WES	*MYH7/* NM_000257.3	p.Arg453Cys/c.1357C>T ^1^ (rs121913625)	14:23429005-G>A	0	Likely Pathog/ Pathog	Pathog (10×), Likely Pathog (1×)	Described [32,33,34]

^1^ Variants verified as de novo. ^2^ Variants also considered as disease-causing de novo candidate mutations but verified during a subsequent family segregation study as of parental origin. Allele frequency - allele frequency in gnomAD, 1000G P3, and ESP6500. WES - whole exome sequencing, TSC - TruSight Cardio; TSO - TruSight One, VUS - variant of uncertain significance, Pathog - pathogenic, ND - no data.

**Table 2 jcm-09-00370-t002:** Molecular characteristics of variants of interest in novel genes from whole exome sequencing (WES) analysis in trios found in probands.

Family			FD05	FD08	FD07	FD01
Gene/ Transcript			TRIB3/ NM_021158.4	SLC2A6/ NM_017585.3	APOBEC3B/ NM_004900.4	UNC45A/ NM_018671.5
Variant			p.Gly257Ser/c.769G>A ^1^	p.Arg283His/c.848G>A ^1^	p.Cys217Tyr/c.650G>A ^2^	p.Arg633Trp/c.1897C>T ^3^
ID			rs534951995	ND	ND	rs374670572
Genomic Coordinates (GRCh38)			20:396382-G>A	9:133475040-C>T	22:38989537-G>A	15:90949334-C>T
ACMG Verdict (default/if *de novo*)			VUS/ Likely Pathog	VUS/ Likely Pathog	VUS/ Likely Pathog	VUS/ VUS
Allele frequency	GnomAD		7.98e–06	0	0	3.94e–04
	1000G P3 Eur/tot		0/2.0e–04	0/0	0/0	0/7.99e–04
	ESP6500 Eur/tot		0/0	0/0	0/0	1.20e–04/ 8.00e–05
Prediction Scores	General	DANN	0.9988	0.9993	0.9458	0.9979
		Mutation Taster	Disease causing	Disease causing	Polymorphism	Disease causing
		FATHMM	Tolerated	Tolerated	Damaging	Tolerated
		Meta SVM	Damaging	Tolerated	Damaging	Tolerated
		MetalR	Tolerated	Tolerated	Damaging	Tolerated
	Conservation	GERP	NR 5.269, RS 5.269	NR 5.3, RS 5.3	NR 1.919, RS 1.919	NR 5.28, RS 4.36
		Mutation Assessor	High	Low	High	Low
	Functional	Provean	Damaging	Neutral	Damaging	Damaging

^1^ Variants verified as de novo. ^2^ Variant in hemizygous state. ^3^ Variant in possible mosaicism. EUR - European; TOT - total, ND - no data, VUS - variant of uncertain significance, Pathog - pathogenic, ND - no data.

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
