# Peer review of "Analysis of De Novo Mutations in Sporadic Cardiomyopathies Emphasizes Their Clinical Relevance and Points to Novel Candidate Genes"

_jcm, 2020, doi:10.3390/jcm9020370_

Round 1

Reviewer 1 Report

The authors should be commended on this interesting study. I offer the following possible revisions and comments for consideration. 

Introduction could be refined. Sporadic cases should not be overgeneralized as acquired disease. In fact, it would be helpful to emphasize that often in pediatric cases there is no family history. The discussion on the meaning/significance of de novo mutations should include mention of effect of such mutations (i.e. they often lead to reduced reproductive fitness). Again in the introduction it would be important to note that many de novo mutations are in syndromic cases which contrast with adult cardiomyopathy which is usually isolated with no extracardiac features. 

For methods, <45 is not necessary considered "young" given the natural history of cardiomyopathy. 

The hypoplastic LV/aorta case seems to confuse the picture. Are the authors sure that there hypoplasia is not somewhere contribution to the cardiomyopathy phenotype?

For the presentation of the results, a summary of the patient characteristics and a detailed table of clinical features and WES findings would improve readability. 

The conclusion regarding the candidate genes and correlation with immune function is overstated given lack of supporting data to functionally validate these findings. 

Author Response

The authors should be commended on this interesting study. I offer the following possible revisions and comments for consideration. 

Introduction could be refined.

Sporadic cases should not be overgeneralized as acquired disease. In fact, it would be helpful to emphasize that often in pediatric cases there is no family history.

Answer: We agree and modified the part of Introduction as follows: “The majority of cardiomyopathies have an autosomal dominant inheritance pattern. Therefore, genetic testing is typically offered to patients with a positive family history whereas sporadic cases may mistakenly be classified as acquired without considering the potential role of de novo mutations. This is particularly important considering paediatric cases with no family history.” (v50-52).

The discussion on the meaning/significance of de novo mutations should include mention of effect of such mutations (i.e. they often lead to reduced reproductive fitness).

Answer: We agree that a broader view of the impact of de novo mutations on human condition will increase the quality of the manuscript and we added to the Discussion: “An important aspect of de novo mutations is the reproductive fitness impairment they cause. This was in particular demonstrated for neurodevelopmental diseases, in which de novo mutations are responsible for the relatively high prevalence of these disorders in the general population (reviewed in [41]). Furthermore, in the X-linked Duchenne dystrophy, the percentage of de novo mutations in affected males reaches up to 40% [42].” (v343-347).

Again in the introduction it would be important to note that many de novo mutations are in syndromic cases which contrast with adult cardiomyopathy which is usually isolated with no extracardiac features. 

Answer: We thank Reviewer for this valuable comment. Indeed, papers published so far confirm this important relationship. We now emphasize it in the Introduction: “It should be noted that the majority of cardiomyopathies causing de novo mutations have been reported in syndromic cases such as those caused by variants of DMD (peripartum cardiomyopathy [3]), DSP (Naxos-Carvajal syndrome [4], erythrokeratodermia-cardiomyopathy syndrome [5]), LAMP2 (Danon disease [6]), PRKAG2 (glycogen storage disease of heart [7,8]), RAF1 (Noonan syndrome [9]), TAZ (Bart syndrome [10]), RRAGC (syndromic foetal dilated cardiomyopathy [11]) and LMNA (atypical progeroid syndrome and dilated cardiomyopathy [12]). Data on de novo mutations in non-syndromic cardiomyopathies are limited and include variants in ACTC [13], MYH7 [14–17], TNNI3 [18], TNNT2 [19,20] and TPM1 [21,22].” (v57-64).

For methods, <45 is not necessary considered "young" given the natural history of cardiomyopathy. 

Answer: We agree with this comment, thus, this phrase has been removed from the manuscript. The sentence is now as follows: “We identified 12 probands with sporadic cardiomyopathy who fulfilled the abovementioned criteria.” (v77-78).

The hypoplastic LV/aorta case seems to confuse the picture. Are the authors sure that there hypoplasia is not somewhere contribution to the cardiomyopathy phenotype?

Answer: We agree with the raised point. However, as we have identified in this proband a known pathogenic mutation in MYH7 which turned out to be de novo we think that the clinical picture in this case, although it may be atypical, is most likely caused by this mutation. In order to better describe the phenotype we removed the phrase “one had restrictive cardiomyopathy (RCM) with hypoplastic left ventricle (LV) and the aorta” from “Materials and Methods” (v83) and rewrote the clinical description in “Results”: “Subsequent imaging studies including echocardiogram, CMR and angio-CT scan revealed left ventricle with reduced diastolic volume and narrow entire aorta, hypertrophied right ventricle, most probably due to pulmonary hypertension.” (v230-232).

For the presentation of the results, a summary of the patient characteristics and a detailed table of clinical features and WES findings would improve readability. 

Answer: We agree that a detailed clinical characteristic of patients is a valuable part of the work. Due to large amount of information resulting in large size of this table, we decided to place it in the Supplementary Materials as “Table S2: The detailed clinical characteristics of probands”.

Data from WES are divided in two sections. In the main manuscript as Table 1. “Molecular characteristics of variants of interest in known cardiomyopathies genes found in probands.” (p6, v235) and Table 2. “Molecular characteristics of variants of interest in novel genes from WES analysis in trios found in probands.” (p10, v333). Also, in the a Supplementary Materials we have added Table S3:  “Other rare (<0,00001 in gnomAD) variants identified in probands as inherited from a parent”.

The conclusion regarding the candidate genes and correlation with immune function is overstated given lack of supporting data to functionally validate these findings. 

Answer: We agree that our Conclusions should be regarded as hypothesis generating and that functional studies will be helpful and valuable. We have modified “Conclusions” as follows: “However, these hypotheses need to be verified by further studies, functional analyses and by identification of additional patients with mutations in respective genes, as the role of the variants we found is difficult to evaluate based only on their de novo origin.” (v438-441). Also we have emphasize this in the “Limitations”: “The lack of functional analyses does not allow to confirm the hypothetical role of the identified genes and their association with the disease.” (v431-432).

Reviewer 2 Report

Authors performed a comprehensive genetic analysis using WES approach in sporadic cases of cardiomyopathies.

Conclusions are in concordance with results.

Only minor points to clarify:

1.- could you perform a table showing technical results of WES (coverage at 20x or 30x, media coverage, exons not covered, etc..) for each sample?

2.- Rare varinat were confirmed using SANGER technology? No indels? What about CNV?

3.- could you add data concerning other rare varinats identified as potential candidate genes responsible for the disease?

After WES, only one rare variant was identified as causative? No additonal ultra-rare variants (MAF<0.001%) in any gene?

Author Response

Authors performed a comprehensive genetic analysis using WES approach in sporadic cases of cardiomyopathies.

Conclusions are in concordance with results.

Only minor points to clarify:

1.- could you perform a table showing technical results of WES (coverage at 20x or 30x, media coverage, exons not covered, etc..) for each sample?

Answer: These important data now can be find in Supplementary Materials Table S1. “Technical parameters of WES in trios”. In the case of uncovered exons we have showed the coverage for probes (baits) in relation to all probes/baits in each WES enrichment in trios.

2.- Rare variant were confirmed using SANGER technology? No indels? What about CNV?

Answer: We confirmed variants found by WES with Sanger sequencing in order to verify their lack in parents and the de novo origin. Please see the following text (v117-136):

“Baseline analysis of NGS results consisted of searching for genetic variants with very low frequency (<0.001) and high bioinformatic pathogenicity prediction scores with special regard to phenotypically plausible genes. The frequencies of variants were derived from the gnomAD (http://gnomad.broadinstitute.org), NHLBI GO Exome Sequencing Project (ESP) 6500 (https://esp.gs.washington.edu/drupal) and an in-house database of >1000 Polish subjects examined by WES. For the bioinformatic prediction scores we used data summarized in VarSome (https://varsome.com). The clinical significance of the variants was based on ClinVar (https://www.ncbi.nlm.nih.gov/clinvar).

If a potentially damaging variant in one of the known genes associated with cardiomyopathies was found in the proband, instead of performing WES for a trio, a more cost-effective approach was used: to confirm or exclude the presence of variants of interest identified by NGS we performed Sanger sequencing in a trio setting (probands-parents) and, if available, in other relatives. For Sanger sequencing we used 3500xL Genetic Analyzer (Life Technologies, Carlsbad, CA, USA) and BigDye Terminator v3.1 Cycle Sequencing Kit (Life Technologies) following the manufacturer’s instructions. Chromatograms were analysed using Variant Reporter 1.1 (Life Technologies). Finding of a de novo mutation in a cardiomyopathy-associated gene ended the diagnostic process.

In the remaining cases, with no candidate variant found using the strategy described above, WES was carried out also for the proband’s parents. Further analysis was performed as described above (including bioinformatic analysis and variant Sanger sequencing) with focus on rare variants present in the proband but absent in the parents.”

The CNVs were also analyzed and the description of this has been added to "Materials and Methods" “Additionally, CNVs (Copy Number Variants)  analysis was performed using the CNVkit [25] with a reference population of around 30 samples.” (v136-137). Also, in the Results (3.2.4) we added information that the CNV variant in APOBEC3B gene was found also with this method (as opposed to the approach based on “single nucleotide variants (SNVs) configuration in the trio”. This sentence is now as follows: “Based on copy number variant analysis confirmed by single nucleotide variants (SNVs) configuration from the trio analysis, we found evidence in the proband of a CNV removing the APOBEC3B gene (apolipoprotein B mRNA editing enzyme catalytic subunit 3B, MIM #607110). The deletion was inherited from the mother, unmasking a hemizygous p.Cys217Tyr/c.650G>A variant of paternal origin (Figure 2).” (v313-317).  No other de novo CNV variants were found. Also, Limitations were modified: “The detection of mutations was limited to the coding sequence and the CNV analysis performed did not allow to detect other types of structural variants such as inversions, balanced translocations or complex rearrangements.”(v428-431).

3.- could you add data concerning other rare variants identified as potential candidate genes responsible for the disease? After WES, only one rare variant was identified as causative? No additional ultra-rare variants (MAF<0.001%) in any gene?

Answer: In the first 8 cases there were 1-2 variants in known cardiomyopathy genes, all were verified with Sanger sequencing and only one was confirmed as de novo. Similarly, for the remaining four cases only one variant in target sequence was found and confirmed as de novo.

We agree that other ultra-rare variants could contribute to the disease in probands and we have added their list in Supplementary Materials Table S3:  “Other rare (<0,00001 in gnomAD) variants identified in probands as inherited from a parent”. In Results (v260-262) we have added information about this data: “Table S3, included in Supplementary Materials, also shows all ultra-rare (<0,00001) variants found in probands that could contribute to the disease but were identified as inherited from parent.”

Round 2

Reviewer 1 Report

To the authors:

Thank you for this revised version. Just a few minor comments:

Line 58: I would reconsider if DMD should be consider syndromic in the context of peripartum cardiomyopathy. I agree in that DMD is associated with Duchenne and Becker muscular dystrophy and a neuromuscular phenotype, but a large proportion of peripartum CM is associated with TTN mutations and I would not consider it "syndromic."

Lines 77-83: Consider moving to results section.

Line 156: Is the recurrent tonsillitis relevant? If not, I would consider removing it.

Line 220: I am assuming this is included because of the association with CRYAB and ocular features (line 364-365)? I am more aware of cataracts as a feature but not myopia? Could you provide a reference?

Author Response

Line 58: I would reconsider if DMD should be consider syndromic in the context of peripartum cardiomyopathy. I agree in that DMD is associated with Duchenne and Becker muscular dystrophy and a neuromuscular phenotype, but a large proportion of peripartum CM is associated with TTN mutations and I would not consider it "syndromic."

Answer: We agree and the phrase regarding DMD has been removed (v58).

Lines 77-83: Consider moving to results section.

Answer: The paragraph describing the study group was moved to the Results section (currently v132-138).

Line 156: Is the recurrent tonsillitis relevant? If not, I would consider removing it.

Answer: We think that the information about recurrent tonsillitis is potentially clinically relevant and thus we would like to retain it since any recurrent infection including tonsillitis/streptococcal pharyngitis can contribute to the development of myocarditis, which is well established risk factor for DCM. In addition, her son also had cardiomyopathy after an infection: “she gave vaginal birth to a male child, in whom DCM was diagnosed at the age of three, following chickenpox” (v157).

Line 220: I am assuming this is included because of the association with CRYAB and ocular features (line 364-365)? I am more aware of cataracts as a feature but not myopia? Could you provide a reference?

Answer: Thank you very much for this comment. Potentially disease-causing (pathogenic/likely pathogenic) variants in the CRYAB gene are rare (only 11 are listed in ClinVar). The CRYAB associated phenotypes are heterogeneous being almost equally divided into cardio-muscular features and cataract. We think that at present it is difficult to determine with certainty the extent of CRYAB pathogenic spectrum and it is possible that this spectrum includes additional phenotypes such as myopia. Given this as well as the fact that little is known about the genetic background of myopia, we think that the coexistence of the CRYAB mutation and myopia in our family is worth mentioning. However, we agree that this may be a too far-reaching speculation, so we rewrote the sentence in the Discussion: “Although it is not clear whether this variant causes disease, it is noteworthy that pathogenic CRYAB variants are associated with cardiomyopathies/myopathies and ocular features such as cataract (https://omim.org). However, given the limited knowledge on both the CRYAB pathogenic spectrum (only 11 variants are listed in ClinVar) and genetics of myopia it is intriguing to speculate that the observed in the family FD10 coexistence of dilated cardiomyopathy and myopia may be caused by the CRYAB variant.” (v363-368).